# Consistent and Truthful Interpretation with Fourier Analysis

## Abstract

For many interdisciplinary fields, ML interpretations need to be consistent with *what-if* scenarios related to the current case, i.e., if one factor changes, how does the model react? Although the attribution methods are supported by the elegant axiomatic systems, they mainly focus on individual inputs and are generally inconsistent. In this paper, we show that such inconsistency is not surprising by proving the impossible trinity theorem, stating that interpretability, consistency, and efficiency cannot hold simultaneously. When consistent interpretation is required, we introduce a new notion called truthfulness, as a relaxation of efficiency. Under the standard polynomial basis, we show that learning the Fourier spectrum is *the unique way* for designing consistent and truthful interpreting algorithms. Experimental results show that for neighborhoods with various radii, our method achieves 2x - 50x lower interpretation error compared with the other methods.

## 1 Introduction

Interpretability is a central problem in deep learning. During training, the neural network strives to minimize the training loss without other distracting objectives. However, to interpret the network, we have to construct a different model[1], which tends to have simpler structures and fewer parameters, e.g., a decision tree or a polynomial. Theoretically, these restricted models cannot perfectly interpret deep networks due to their limited representation power. Therefore, the previous researchers had to introduce various relaxations. The most popular and elegant direction is the attribution methods with axiomatic systems (Sundararajan et al., 2017; Lundberg & Lee, 2017), which mainly focus on individual inputs.

The interpretations of the attribution methods do not automatically extend to the neighboring points. Take SHAP (Lundberg & Lee, 2017) as the motivating example, illustrated in Figure 1 on the task of sentiment analysis of movie reviews. In this example, the interpretations of the two slightly different sentences are not consistent. It is not only because the weights of each word are significantly different but also because, after removing a word "very" of weight $19.9\%$, the network's output only drops by $97.8\% - 88.7\% = 9.1\%$. In other words, **the interpretation does not explain the network's behavior even in a small neighborhood of the input**.

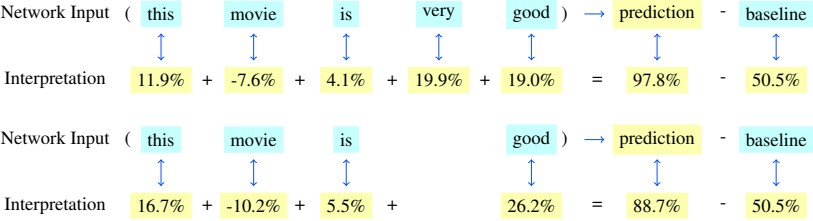

Figure 1: Interpretations generated by SHAP on a movie review.

Inconsistency is not a vacuous concern. Imagine a doctor treating a diabetic with the help of an AI system. The patient has features A, B, and C, representing three positive signals from various tests.

---

[1]For simplicity, below we use **model** to denote the model that provides interpretation and **network** to denote the general black-box machine learning model that needs interpretation.

AI recommends giving $4$ units of insulin with the following explanation: A, B, and C have weights $1$, $1$, and $2$, respectively, so $4$ units in total. Then the doctor may ask AI: *what if* the patient only has A and B, but not C? One may expect the answer to be close to 2, as A+B has a weight of 2. However, the network is highly non-linear and may output other suggestions like 3 units, explaining that both A and B have a weight of $1.5$. Such inconsistent behaviors will drastically reduce the doctor's confidence in the interpretations, limiting the AI system's practical value.

Consistency (see Definition 3) is certainly not the only objective for interpretability. Being equally important, efficiency is a commonly used axiom in the attribution methods (Weber, 1988; Friedman & Moulin, 1999; Sundararajan & Najmi, 2020), also called local accuracy (Lundberg & Lee, 2017) or completeness (Sundararajan et al., 2017), stating that the model's output should be equal to the network's output for the given input (see Definition 4). Naturally, one may ask the following question:

> **Q1: Can we always generate an interpreting model that is both efficient and consistent?**

Unfortunately, this is generally impossible. We have proved the following theorem in Section 3:

**Theorem 1**(Impossible trinity, informal version). *Interpretability, consistency, and efficiency cannot hold simultaneously.*

A few examples following Theorem 1:

(a) Attribution methods are interpretable and efficient, but not consistent.

(b) The original (deep) network is consistent and efficient, but not interpretable.

(c) If one model is interpretable and consistent, it cannot be efficient.

However, consistency is necessary for many scenarios, so one may have the follow up question:

> **Q2: For consistent interpreting models, can they be approximately efficient?**

The answer to this question depends on how "approximately efficient" is defined. We introduce a new notion called truthfulness, which can be seen as a natural relaxation of efficiency, i.e., partial efficiency. Indeed, we split the functional space of $f$ into two subspaces, the readable part and the unreadable part (see Definition 5). We call $g$ is truthful if it can truthfully represent the readable part of $f$. Notice that the unreadable part is not simply a "fitting error". Instead, it truthfully represents the higher-order non-linearities in the network that our interpretation model $g$, even doing its best, cannot cover. In short, what $g$ tells is true, although it may not cover all the truth. Due to Theorem 1, **this is essentially the best that consistent algorithms can achieve.**

Truthfulness is a parameterized notion, which depends on the choice of the readable subspace. While theoretically there are infinitely many possible choices of subspaces, in this paper we follow the previous researchers on interpretability with non-linearities (Sundararajan et al., 2020; Masoomi et al., 2022; Tsang et al., 2020), and uses the basis that automatically induces interpretable terms like $x_i x_j$ or $x_i x_j x_k$, which capture higher order correlations and are easy to understand. The resulting subspace has the standard polynomial basis (or equivalently, the Fourier basis). Following the notion of consistency and truthfulness, our last question is:

> **Q3: Can we design consistent and truthful interpreting models with polynomial basis?**

It turns out that, when truthfulness is parameterized with the polynomial basis, designing consistent and truthful interpreting models is equivalent to learning the Fourier spectrum (see Lemma 1). In other words, this is **the unique way** to generate truthful and consistent interpretations for high order correlations among input parameters.

In this paper, we focus on the case that $f$ and $g$ are Boolean functions, i.e., the input variables are binary. This is a commonly used assumption in the literature (LIME (Ribeiro et al., 2016), SHAP (Lundberg & Lee, 2017), Shapley Taylor (Sundararajan et al., 2020), and other methods (Sundararajan & Najmi, 2020; Zhang et al., 2021; Frye et al., 2020; Covert et al., 2020; Tsai et al., 2022)), although sometimes not explicitly stated. For readers not familiar with Boolean functions, we remark that Boolean functions have very strong representation power. For example, empirically most human readable interpretations can be converted into (ensemble of) decision trees, and theoretically all

(ensemble of) decision trees can be converted into Boolean functions. The widely used algorithm XGBoost (Chen & Guestrin, 2016) is based on an ensemble of decision trees.

Given any Boolean function, we may expand it on the Fourier basis and get its Fourier spectrum. It is well known that the Fourier spectrum's complexity naturally characterizes the complexity of Boolean functions (O'Donnell, 2014), e.g., a small decision tree can be approximated with a low degree sparse polynomial on the Fourier basis (Mansour, 1994). Therefore, we restrict $g$ functions on its Fourier spectrum. For learning truthful models, we apply two different methods from Boolean functional anslysis, Harmonica (Hazan et al., 2018) and Low-degree (Linial et al., 1993). Both algorithms have rigorous theoretical guarantees on recovery performance and sampling complexities. Afterwards, in Section 5, we will demonstrate that on datasets like SST-2 and IMDb, Harmonica can get 2x-50x lower interpretation error compared with other methods.

In summary, our contributions are:

- We have proved the impossible trinity theorem for interpretability, which shows that interpretable algorithms cannot be consistent and efficient at the same time.

- For interpretable algorithms that are consistent but not efficient, we have introduced a new notion called truthfulness, which can be seen as partial efficiency. Due to the impossible trinity theorem, this is the best that one can achieve when consistency is required.

- For truthfulness with the polynomial basis, we have proved that the problem is equivalent to learning Boolean functions, and empirically demonstrated that the Harmonica algorithm can achieve much better interpretation errors and truthfulness, compared with the other methods.

## 2 RELATED WORK

Interpretability is a critical topic in machine learning, and we refer the reader to (Doran et al., 2017; Lipton, 2018) for insightful general discussions. Below we discuss different types of interpretable models.

**Model-specific interpretable models**   Interpretable/white-box models are inherently ante-hoc and model-specific. One of the goals behind using interpretable models is to have inherent model interpretability. Current mainstream approaches include: Decision Trees (Wang et al. (2015a), Balestriero (2017), Yang et al. (2018)), Decision Rules (Wang et al. (2015b), Su et al. (2015)), Decision Sets (Lakkaraju et al. (2019), Wang et al. (2017)) and Linear Model (Ustun & Rudin (2014), Ustun et al. (2014)).

**Shapley value based explanations**   Shapley value (Shapley, 1953) was first introduced in cooperative game theory, with several strong axiomatic theoretical properties (Weber, 1988; Grabisch & Roubens, 1999). Recently, it has been adopted for explanations of machine learning models (Lundberg & Lee, 2017; Štrumbelj & Kononenko, 2014; Sundararajan & Najmi, 2020; Wang et al., 2021a; Zhang et al., 2021; Frye et al., 2020; Yuan et al., 2021) or feature importance (Covert et al., 2020).

Based on the Shapley value, Owen (1972) proposed the Shapley interaction value to study pairwise interactions between players. Grabisch & Roubens (1999) generalized it to study interactions of higher orders and provided an axiomatic foundation. Starting from that, many researchers worked on higher-order feature interactions from different perspectives (Sundararajan et al., 2020; Masoomi et al., 2022; Tsang et al., 2020; Aas et al., 2021; Tsai et al., 2022).

However, computing Shapley value is NP-hard (Elkind et al., 2009; Van den Broeck et al., 2022). Recently many works on Shapley value focus on how to approximate Shapley value efficiently, including sampling-based methods, regression-based methods, or learning-based methods (Lundberg & Lee, 2017; Chen et al., 2019; Ancona et al., 2019; Covert & Lee, 2021; Jethani et al., 2021; Hamilton et al., 2022; Wang et al., 2021b).

**Gradient-based explanations**   Simonyan et al. (2014) proposed a method to approximate the network and propose the gradient as attribution linearly. Smoothgrad (Smilkov et al., 2017) generates explanations by introducing perturbation to the input and then observing the corresponding effect

on the model's predictions. Integrated Gradients (Sundararajan et al., 2017) distributes the change in output with respect to a baseline input by integrating gradients between the two input states, and Janizek et al. (2021) generalized this method to second order.

DeepLIFT (Shrikumar et al., 2017) and LRP (Montavon et al., 2017) backpropagated contribution layer-wise to every feature of the input. Grad-CAM (Selvaraju et al., 2017) leverage activation values of convolutional layers. CXPlain (Schwab & Karlen, 2019) removes a single feature from a single input and measures the change in the loss function.

**Consistent methods** LIME (Ribeiro et al., 2016) is a classical method that samples the data points following a predefined sampling distribution and computes a function that empirically satisfies local fidelity. Chen et al. (2018) uses a two-layer additive risk model for interpreting the credit risk assessment. Bastani et al. (2017) proposes an approach called model extraction via greedily learning a decision tree that approximates $f$. However, these methods are mainly designed with heuristics and do not have any theoretical guarantees.

**Other methods** Covert et al. (2021) introduced a general removal-based framework that unifies 26 interpretability algorithms, including classical methods like DeepLIFT (Shrikumar et al., 2017), LRP (Montavon et al., 2017), etc.

## 3 OUR FRAMEWORK ON INTERPRETABILITY

We consider a Hilbert space $\mathcal{H}$ equipped with inner product $\langle \cdot \rangle$, and induced $\ell_2$ norm $\|\cdot\|$. We denote the input space by $\mathcal{X}$, the output space by $\mathcal{Y}$, which means $\mathcal{H} \subseteq \mathcal{X} \to \mathcal{Y}$. We focus on Boolean functions in this paper, so $\mathcal{X} = \{-1, 1\}^n, \mathcal{Y} = \mathbb{R}$. We use $-1/1$ instead of $0/1$ to represent the binary variables because it easily fits into the Fourier basis. Due to the space limit, we defer a brief introduction on Fourier analysis to Appendix A. Fourier analysis of Boolean function is a fascinating field, and we refer the reader to O'Donnell (2014) for a more comprehensive introduction.

We use $\mathcal{G} \subset \mathcal{H}$ to denote the set of interpretable functions, and $\mathcal{F} \subset \mathcal{H}$ to denote the set of machine learning models that need interpretation. In this paper, we focus on models that are not self-interpretable, i.e., $f \in \mathcal{F} \setminus \mathcal{G}$.

**Definition 1** (Interpretable). *A model $g$ is interpretable, if $g \in \mathcal{G}$.*

Interpretable models are generated by interpretation algorithms.

**Definition 2** (Interpretation Algorithm). *An interpretation algorithm $\mathcal{A}$ takes $f \in \mathcal{H}, x \in \mathcal{X}$ as inputs, and outputs $\mathcal{A}(f, x) \in \mathcal{G}$ for interpreting $f$ on $x$.*

As we mentioned previously, for many interdisciplinary fields, the interpretation algorithm should be consistent.

**Definition 3** (Consistent). *Given $f \in \mathcal{H}$, an interpretation algorithm $\mathcal{A}$ is consistent with respect to $f$, if $\mathcal{A}(f, x)$ is same for every $x \in \mathcal{X}$.*

Efficiency is an important property of the attribution methods.

**Definition 4** (Efficient). *A model $g \in \mathcal{H}$ is efficient with respect to $f \in \mathcal{F}$ on $x \in \mathcal{X}$, if $g(x) = f(x)$.*

The following theorem states that one cannot expect to achieve the best of all three worlds.

**Theorem 1** (Impossible trinity). *For any interpretation algorithm $\mathcal{A}$ and function sets $\mathcal{G} \subset \mathcal{F} \subseteq \mathcal{H}$, there exists $f \in \mathcal{F}$ such that with respect to $f$, either $\mathcal{A}$ is not consistent, or $\mathcal{A}(f, x)$ is not efficient on $x$ for some $x \in \mathcal{X}$.*

*Proof.* Pick $f \in \mathcal{F} \setminus \mathcal{G}$. If $\mathcal{A}$ is consistent with respect to $f$, let $g = \mathcal{A}(f, x) \in \mathcal{G}$ for any $x \in \mathcal{X}$. If for every $x \in \mathcal{X}, g(x) = f(x)$, we know $g = f \notin \mathcal{G}$, this is a contradiction. Therefore, there exists $x \in \mathcal{X}$ such that $g(x) \neq f(x)$. □

Theorem 1 says efficiency is too restrictive for consistent interpretations. However, being inefficient does not mean the interpretation is wrong, it can still be truthful. Recall a subspace $V \subset \mathcal{H}$ is *closed* if whenever $\{f_n\} \subset V$ converges to some $f \in \mathcal{H}$, then $f \in V$. We have:

**Definition 5** (Truthful gap and truthful). *Given a closed subspace $V \subseteq \mathcal{H}$, $g \in \mathcal{G} \subseteq V$ and $f \in \mathcal{F}$, the truthful gap of $g$ to $f$ for $V$ is:*

$$\mathbb{T}_V(f, g) = \|f - g\|^2 - \inf_{v \in V} \|f - v\|^2 \tag{1}$$

*When $\mathbb{T}_V(f, g) = 0$, we say $g$ is* truthful *for subspace $V$ with respect to $f$, and we know (see e.g. Lemma 4.1 in Stein & Shakarchi (2009)) $\forall v \in V, \langle f - g, v \rangle = 0$.*

Truthfulness means $g$ fully captures the information in the subspace $V$ of $f$, therefore it can be seen as a natural relaxation of efficiency. To characterize the interpretation quality, we introduce the following notion.

**Definition 6** (Interpretation error). *Given functions $f, g \in \mathcal{X} \to \mathcal{Y}$, the interpretation error between $f$ and $g$ with respect to measure $\mu$ is*

$$\mathbb{I}_{p,\mu}(f, g) = \left( \int_{\mathcal{X}} |f(x) - g(x)|^p d\mu(x) \right)^{1/p} \tag{2}$$

Notice that interpretation error is only a *loss function* that measures the quality of the interpretation, instead of a metric in $\ell_p$ space. Therefore, $\mu$ can be a non-uniform weight distribution following the data distribution. For real-world applications, interpreting the model over the whole $\mathcal{X}$ is unnecessary, so $\mu$ is usually defined as a uniform distribution on the neighborhood of input $x$ (under a certain metric), in which case we denote the distribution as $\mathcal{N}_x$.

Other than loss functions defined over the whole function space, sometimes we also need the notion for pointwise interpretation error.

**Definition 7** (Pointwise interpretation error). *Given two functions $f, g \in \mathcal{H}$, an input $x$, the interpretation error of $g$ to $f$ on $x$ is $u(x) = |f(x) - g(x)|$.*

**Discussion on universal consistency.** When talking about the notion of consistency, there are two entirely different settings, which we name as the "global consistency" and the "universal consistency". The global consistency is what we focus in this paper (Definition 3), i.e., the interpretation only relates to the input features but not the others. This scenario belongs to the category of "removal-based explanations", and almost all the existing interpretable methods also belong to this category (26 of them are discussed in Covert et al. (2021)). On the other hand, the universal consistency means the interpretation may depend on the features that are different from the input features. Think about interpreting the sentence "I am happy" with the interpretation that "this sentence does not include [very], so this guy is not super happy". Universal consistency is much more challenging than global consistency, and we suspect more powerful machinery like foundation models are needed, which we leave as future work.

## 4 LEARNING BOOLEAN FUNCTIONS

With the Fourier basis, we define our interpretable function set $\mathcal{G}$ as follows.

**Definition 8** ($C$-Readable function). *Given a set of Fourier bases $C$, a function $f$ is $C$-readable if it is supported on $C$. That is, for any $\chi_S \notin C$, $\langle f, \chi_S \rangle = 0$. Denote the corresponding subspace as $V_C$.*

The Readable notion is parameterized with $C$, because it may differ case by case. If we set $C$ to be all the single variable bases, only linear functions are readable; if we set $C$ to be all the bases with the degree at most 2, functions with pairwise interactions are also readable. Moreover, if we further add one higher order term to $C$, e.g., $\chi_{\{x_1,x_2,x_3,x_4\}}$, it means we can also reason about the factor $x_1 x_2 x_3 x_4$ in the interpretation, which might be an important empirical factor that people can easily understand. Starting from the bases set $C$, we have the following formula for computing the truthful gap.

**Lemma 1** (Truthful gap for Boolean functions). *Given a set of Fourier bases $C$, two functions $f, g \in \{-1, 1\}^n \to \mathbb{R}$, the truthful gap of $g$ to $f$ for $C$ is*

$$\mathbb{T}_{V_C}(f, g) = \sum_{\chi_S \in C} \langle f - g, \chi_S \rangle^2 \tag{3}$$

*Proof.* Denote the complement space as $V_{\bar{C}}$. We may expand $f, g, v$ on both bases, and get:

$$\|f - g\|^2 - \inf_{v \in V} \|f - v\|_2$$

$$= \sum_{S \in C} \langle f - g, \chi_S \rangle^2 + \sum_{S \in \bar{C}} \langle f, \chi_S \rangle^2 - \inf_{v \in V_C} \left( \sum_{S \in C} \langle f - v, \chi_S \rangle^2 + \sum_{S \in \bar{C}} \langle f, \chi_S \rangle^2 \right)$$

$$= \sum_{S \in C} \langle f - g, \chi_S \rangle^2 - \inf_{v \in V_C} \left( \sum_{S \in C} \langle f - v, \chi_S \rangle^2 \right) = \sum_{S \in C} \langle f - g, \chi_S \rangle^2$$

Where the last equality holds because we can set the Fourier coefficients $\hat{v}_S = \hat{f}_S$ for every $S \in C$, which further gives $\langle f - v, \chi_S \rangle = 0$. □

With the previous definitions, it becomes clear that finding a truthful interpretation $g$ is equivalent to accurately learning a Boolean function with respect to the readable bases set $C$. Intuitively, it means we want to find algorithms that can compute the coefficients for the bases in $C$. In other words, we want to find the importance of the bases like $x_1, x_2 x_5, x_2 x_6 x_7$, etc. Learning Boolean function is a classical problem in learning theory, and there are many algorithms like KM algorithm (Kushilevitz & Mansour, 1991), Low-degree algorithm (Linial et al., 1993) and Harmonica (Hazan et al., 2018).

We pick two algorithms for our task: Harmonica and Low-degree. Compared with Low-degree, Harmonica has much better sampling efficiency based on compressed sensing techniques. Specifically, for general real-valued decision trees which has $s$ leaf nodes and are bounded by $B$, the sample complexity of Harmonica is $\tilde{O}\left(B^2 s^2 / \varepsilon \cdot \log n\right)$, while Low-degree is $\tilde{O}\left(B^4 s^2 / \varepsilon^2 \cdot \log n\right)$ (Hazan et al., 2018). However, the Low-degree algorithm works for more general settings, while the theoretical guarantee of Harmonica depends on the assumption that $f$ is approximately sparse in the Fourier space. Therefore, we include both algorithms for completeness. Empirically, Harmonica has a much better performance than Low-degree.

We defer the Algorithm description and theoretical guarantees to Appendix B and Appendix C, and discussion on the comparison of our algorithms with the existing algorithms to Appendix D.

**Remarks**. The theoretical guarantee of Harmonica assumes the target function $f$ is approximately sparse in the Fourier space, which means most of the energy of the function is concentrated in the bases in $C$. This is not a strong assumption, because if $f$ is not approximately sparse, it means $f$ has energy in many different bases, more or specifically, the bases with higher orders. In other words, $f$ has a large variance and is difficult to interpret. In this case, no existing algorithms will be able to give consistent and meaningful interpretations.

Likewise, although the Low-degree algorithm does not assume sparsity for $f$, it cannot learn all possible functions accurately as well. There are $2^n$ different bases, and if we want to learn the coefficients for all of them, the cumulative error for $g$ is at the order of $\Omega(2^n \epsilon)$, which is exponentially large. This is not surprising due to the no free lunch theorem in the generalization theory, as we do not expect to be able to learn "any functions" without exponentially many samples.

## 5 EXPERIMENTS

### 5.1 ANALYSIS ON THE POLYNOMIAL FUNCTIONS

To investigate the performance of different interpretation methods, we *manually* investigate the output of different algorithms (LIME (Ribeiro et al., 2016), SHAP (Lundberg & Lee, 2017), Shapley Interaction Index(Owen, 1972), Shapley Taylor (Sundararajan et al., 2020; Hamilton et al., 2022), Faith-SHAP (Tsai et al., 2022), Harmonica and Low-degree) for lower order polynomial functions.

We observe that all algorithms can accurately learn the coefficients of the first order polynomial. For the second order polynomial function, only Shapley Taylor, Faith-SHAP, Harmonica and Low-degree can learn all the coefficients accurately. For the third order polynomial function, only Faith-SHAP, Harmonica and Low-degree succeeded. Due to the space limit, we defer the details to Appendix E.

Faith-SHAP has a delicate representation theorem, which assigns coefficients to different terms under the Möbius transform. Since the basis induced by the Möbius transform is not orthonormal, it is

not clear to us whether Faith-SHAP can theoretically compute the accurate coefficients for higher order functions. However, the running time of Faith-SHAP has exponential dependency on $n$, so empirically weighted sampling on subsets of features is needed Tsai et al. (2022). This might be the main reason that our algorithms outperform Faith-SHAP in the experiments with real datasets.

## 5.2 Experimental setup

In the rest of this section, we conduct experiments to evaluate the interpretation error $\mathbb{I}_{p,\mathcal{N}_x}(f,g)$ and truthful gap $\mathbb{T}_{V_C}(f,g)$ of Harmonica and other baseline algorithms on NLP and vision tasks quantitatively. In our experiments, we choose 2nd order and 3rd order Harmonica algorithms, which correspond to setting $C$ to be all terms with the order at most 2 and 3. The baseline algorithms chosen for comparison include LIME, Integrated Gradients (Sundararajan et al., 2017), SHAP, Integrated Hessians (Janizek et al., 2021), Shapley Taylor interaction index, and Faith-Shap where the first three are first-order algorithms, and the last three are second-order algorithms.

The two language tasks we pick are the SST-2 (Socher et al., 2013) dataset for sentiment analysis and the IMDb (Maas et al., 2011) dataset for movie review classification. And the vision task is the ImageNet (Krizhevsky et al., 2012) for image classification.

## 5.3 Sentiment Analysis

We start with a binary sentiment classification task on SST-2 dataset. We try to interpret a convolutional neural network (see details in Appendix F) trained with Adam (Kingma & Ba, 2015) optimizer for 10 epochs. The model has a test accuracy of $80.6\%$. For a given input sentence $x$ with length $l$, we define the induced neighborhood $\mathcal{N}_x$ by introducing a masking operation on this sentence. The radius $0 \leq r \leq l$ is defined as the maximum number of masked words.

**Results on interpretation error** Figure 2 shows the interpretation error evaluated under different neighborhoods with a radius ranging from 1 to $\infty$. Here $\infty$ represents the maximum sentence length, which may vary for different data points. The interpretation error is evaluated under $L^2$, $L^1$, and $L^0$ norms. Here $L^2$ and $L^1$ are defined according to Eqn.(2) with $p = 2$ and $p = 1$, respectively. And $L^0$ denotes $\int_{\mathcal{X}} \mathbb{1}\{|f(x) - g(x)| \geq 0.1\} d\mu(x)$. The detailed numerical results are presented in Table 4 in Appendix G. We can see that Harmonica consistently outperforms all the other baselines on all radii.

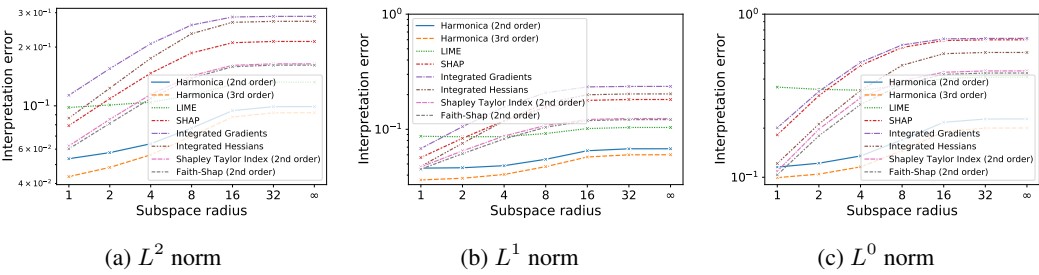

(a) $L^2$ norm          (b) $L^1$ norm          (c) $L^0$ norm

Figure 2: Visualization of interpretation error $\mathbb{I}_{p,\mathcal{N}_x}(f,g)$ evaluated on SST-2 dataset.

**Estimating truthful gap** For convenience, we define the set of bases $C^d$ up to degree $d$ as $C^d = \{\chi_S | S \subseteq [n], |S| \leq d\}$. We evaluate the truthful gap on the set of bases $C^3$, $C^2$, and $C^1$. By definition in Eqn.(3), we have

$$\mathbb{T}_{V_C}(f,g) = \sum_{\chi_S \in C} \langle f - g, \chi_S \rangle^2 = \left( \mathop{\mathbb{E}}_{x \sim \{-1,1\}^n} \left[ (f(x) - g(x)) \sum_{\chi_S \in C} \chi_S(x) \right] \right)^2 .$$

Then we perform a sampling-based estimation of the truthful gap. Worth mentioning that since the size of set $C^d$ satisfies $|C^d| = \sum_{i=0}^{d} \binom{n}{i}$ and the max length of all sentences $n^* = 50$, $\sum_{\chi_S \in C} \chi_S(x)$, as the summation function of orthonormal basis, is easy to compute on every sample $x \in \{-1,1\}^n$ (for $n^*$ very large, we will perform another sampling step on this function).

**Results on truthful gap**  Figure 3 shows the truthful gap evaluated on SST-2 dataset. We can see that Harmonica achieves the best performance for $C^2$ and $C^3$. For the simple linear case $C^1$, Harmonica is almost as good as LIME.

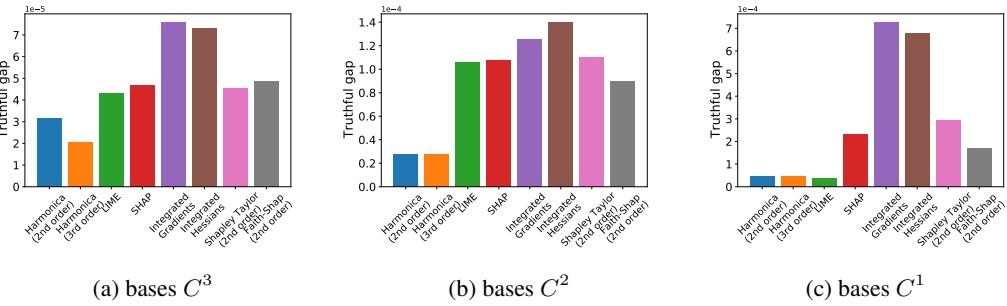

Figure 3: Visualization of truthful gap $\mathbb{T}_C(f, g)$ evaluated on SST-2 dataset.

## 5.4 MOVIE REVIEW CLASSIFICATION

The IMDb dataset contains long paragraphs, and each paragraph has many sentences. For the readability of results, we treat sentences as units instead of words – masking several words in a sentence may make the whole paragraph hard to understand and even meaningless, while masking a critical sentence has meaningful semantic effects. Therefore, the radius is defined as the maximum number of masked sentences. By default, we use periods, colons, and exclamations to separate sentences. The target network to be interpreted is a convolutional neural network (see details in Appendix F) trained over this dataset with an accuracy of $85.6\%$.

**Results on interpretation error**  Figure 4 shows the interpretation error evaluated on IMDb dataset under the same settings of SST-2 dataset, i.e., sampling-based estimation, with a slight modification of this procedure that on IMDb dataset, masking operation is performed on sentences in one input paragraph instead (we also change the definition of radii accordingly). We can see that Harmonica consistently outperforms all the other baselines on all radii. The detailed numerical results are presented in Table 5 in Appendix G.

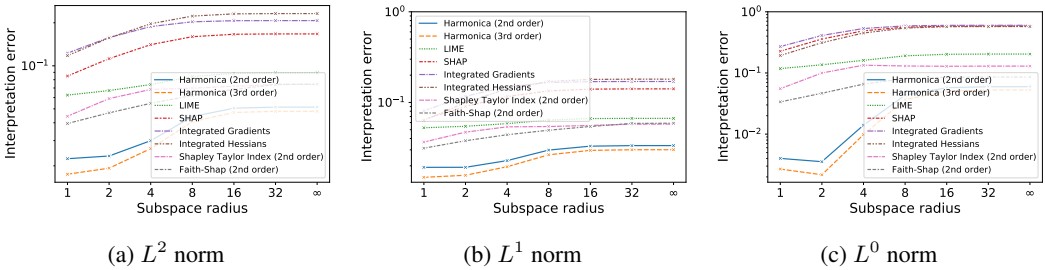

Figure 4: Visualization of interpretation error $\mathbb{I}_{p,\mathcal{N}_x}(f, g)$ evaluated on IMDb dataset.

**Results on truthful gap**  Figure 5 shows the truthful gap evaluated on IMDb dataset under the same settings of SST-2 dataset. We can see that Harmonica consistently outperforms all the other baselines.

## 5.5 IMAGE CLASSIFICATION

We use ImageNet (Krizhevsky et al., 2012) dataset as the vision interpretation dataset. For this task, we aim to provide class-specific interpretation, which means that only the class with the maximum predicted probability is taken into consideration for each sample. Following LIME (Ribeiro et al., 2016), we split each $224 \times 224$ image into 16 superpixels to balance the human readability and computational efficiency. The official pre-trained ResNet-101 (He et al., 2016) model from PyTorch is used.

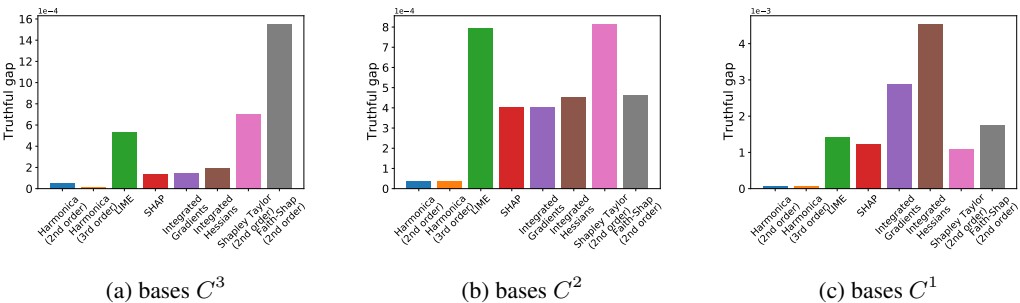

Figure 5: Visualization of truthful gap $\mathbb{T}_C(f, g)$ evaluated on IMDb dataset.

**Results on interpretation error**  Figure 6 shows the interpretation error evaluated on 1000 random images from ImageNet while the masking operation is performed on 16 superpixels in one input image instead (we also change the definition of radii accordingly). We can see that when the neighborhood's radius is greater than 1, Harmonica outperforms all the other baselines. The detailed numerical results are presented in Table 6 in Appendix G.

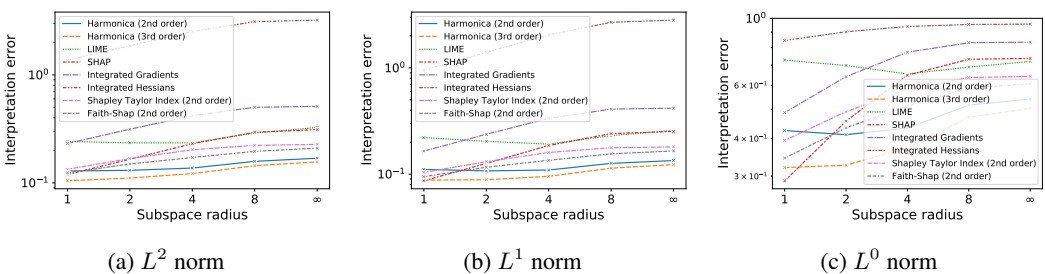

Figure 6: Visualization of interpretation error $\mathbb{I}_{p, \mathcal{N}_x}(f, g)$ evaluated on ImageNet dataset.

**Results on truthful gap**  Figure 7 shows the truthful gap evaluated on ImageNet dataset. We can see that Harmonica outperforms all the other baselines consistently.

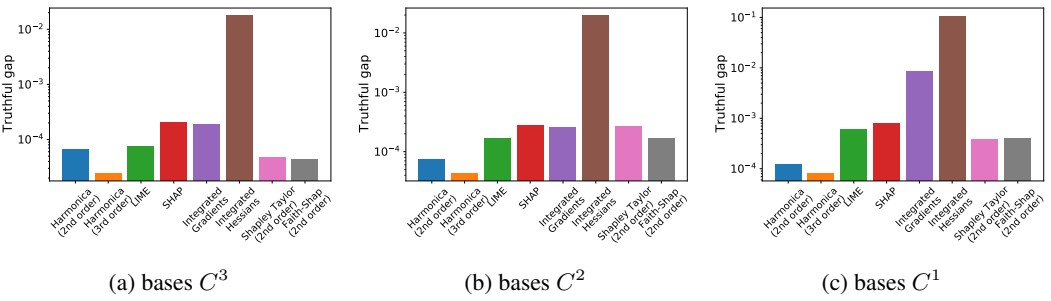

Figure 7: Visualization of truthful gap $\mathbb{T}_C(f, g)$ evaluated on ImageNet dataset.

## 5.6 ADDITIONAL EXPERIMENTS

We further explore the sample complexity of Harmonica and Low-degree algorithms in Appendix H, which show that Harmonica achieves better performance with the same sample size.

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

## A  PRELIMINARIES ON FOURIER ANALYSIS

We first introduce the Fourier basis:

**Definition 9** (Fourier basis). *For any subset of variables $S \subseteq [n]$, we define the corresponding Fourier basis as*

$$\chi_S(x) = \Pi_{i \in S} x_i \ \in \{-1, 1\}^n \to \{-1, 1\}$$

The Fourier basis is a complete orthonormal basis for Boolean functions, under the uniform distribution on $\{-1, 1\}^n$. We remark that this uniform distribution is used for theoretical analysis and algorithm design, and is different from the measure $\mu$ for interpretation quality assessment in Definition 6. We define the inner product as follows.

**Definition 10** (Inner product). *Given two functions $f, g \in \{-1, 1\}^n \to \mathbb{R}$, their inner product is:*

$$\langle f, g \rangle = 2^{-n} \sum_{x \in \{-1,1\}^n} f(x)g(x) = \mathop{\mathbb{E}}_{x \sim \{-1,1\}^n} [f(x)g(x)]$$

Then we can compute the Fourier spectrum of any Boolean functions based on the Fourier basis.

**Definition 11** (Fourier expansion). *For any Boolean function $f \in \{-1, 1\}^n \to \mathbb{R}$, we can expand it as*

$$f(x) = \sum_{S \subseteq [n]} \hat{f}_S \chi_S(x),$$

*where $\hat{f}_S = \langle f, \chi_S \rangle$ is the Fourier coefficient on $S$. All the Fourier coefficients together are called the Fourier spectrum of $f$.*

The inner product defines one kind of similarity between two functions and is invariant under different basis. Specifically, we have the following Theorem.

**Definition 12** (Plancherel's Theorem). *Given two functions $f, g \in \{-1, 1\}^n \to \mathbb{R}$,*

$$\langle f, g \rangle = \sum_{S \subseteq [n]} \hat{f}_S \hat{g}_S$$

When setting $f = g$, we get the Parseval's identity: $\mathbb{E}[f^2] = \sum_S \hat{f}_S^2$.

## B  HARMONICA ALGORITHM

---

**Algorithm 1** Harmonica

---

1. Given uniformly randomly sampled $x_1, \cdots, x_T$, evaluate them on $f$: $\{f(x_1), ...., f(x_T)\}$.
2. Solve the following regularized regression problem.

$$\underset{\alpha \in \mathbb{R}^{|C|}}{\operatorname{argmin}} \left\{ \sum_{i=1}^{T} \left( \sum_{S, \chi_S \in C} \alpha_S \chi_S(x_i) - f(x_i) \right)^2 + \lambda \|\alpha\|_1 \right\} \qquad (4)$$

3. Output the polynomial $g(x) = \sum_{S, \chi_S \in C} \alpha_S \chi_S(x)$.

---

To present the theoretical guarantees of the Harmonica algorithm, we introduce the following definition, which is slightly different from its original version in Hazan et al. (2018).

**Definition 13** (Approximately sparse function). *We say a function $f \in \{-1, 1\}^n \to \mathbb{R}$ is $(\epsilon, s, C)$-bounded, if $\mathbb{E}[(f - \sum_{\chi_S \in C} \hat{f}(S)\chi_S)^2] \leq \epsilon$ and $\sum_S |\hat{f}(S)| \leq s$.*

Here $f$ is $(\epsilon, s, C)$-bounded means it is almost readable and has bounded $\ell_1$ norm. Our algorithm is slightly different from the original algorithm proposed by Hazan et al. (2018), but similar theoretical guarantees still hold, as stated below.

**Theorem 2.** *Given $f \in \{-1, 1\}^n \to \mathbb{R}$ a $(\epsilon/4, s, C)$-bounded function, Algorithm 1 finds a function $g$ with interpretation error at most $\epsilon$ in time $O((T \log \frac{1}{\epsilon} + |C|/\epsilon) \cdot |C|)$ and sample complexity $T = \tilde{O}(s^2/\epsilon \cdot \log |C|)$.*

Our proof is similar to the one in the original paper Hazan et al. (2018), with changes in the readable notion, which is now more flexible than being low order. First recall the classical Chebyshev inequality.

**Theorem 3** (Multidimensional Chebyshev inequality). *Let $X$ be an $m$ dimensional random vector, with expected value $\mu = \mathbb{E}[X]$, and covariance matrix $V = \mathbb{E}\left[(X - \mu)(X - \mu)^T\right]$. If $V$ is a positive definite matrix, for any real number $\delta > 0$ :*

$$\mathbb{P}\left(\sqrt{(X - \mu)^T V^{-1}(X - \mu)} > \delta\right) \leq \frac{m}{\delta^2}$$

*Proof of Theorem 2.* Let $f$ be an $(\varepsilon/4, s, C)$-bounded function written in the orthonormal basis as $\sum_S \hat{f}(S)\chi_S$. We can equivalently write $f$ as $f = h + g$, where $h$ is supported on $C$ that only only includes coefficients of magnitude at least $\varepsilon/4s$ and the constant term of the polynomial expansion of $f$.

Since $L_1(f) = \sum_S \left|\hat{f}_S\right| \leq s$, we know $h$ is $4s^2/\varepsilon + 1$ sparse. The function $g$ is thus the sum of the remaining $\hat{f}(S)\chi_S$ terms not included in $h$. Denote the set of bases that appear in $C$ but not in $g$ as $R$, so we know the coefficient of $f$ on the bases in $R$ is at most $\epsilon/4s$.

Draw $m$ (to be chosen later) random labeled examples $\left\{\left(z^1, y^1\right), \ldots, \left(z^m, y^m\right)\right\}$ and enumerate all $N = |C|$ basis functions $\chi_S \in C$ as $\{\chi_1, \ldots, \chi_N\}$. Form matrix $A$ such that $A_{ij} = \chi_j\left(z^i\right)$ and consider the problem of recovering $4s^2/\varepsilon + 1$ sparse $x$ given $Ax + e = y$ where $x$ is the vector of coefficients of $h$, the $i$ th entry of $y$ equals $y^i$, and $e_i = g\left(z^i\right)$.

We will prove that with constant probability over the choice $m$ random examples, $\|e\|_2 \leq \sqrt{\varepsilon m}$. Applying Theorem 5 in Hazan et al. (2018) by setting $\eta = \sqrt{\varepsilon}$ and observing that $\sigma_{4s^2/\varepsilon+1}(x)_1 = 0$ (see definition in the theorem), we will recover $x'$ such that $\|x - x'\|_2^2 \leq c_2^2 \varepsilon$ for some constant $c_2$. As such, for the function $\tilde{f} = \sum_{i=1}^N x_i' \chi_i$ we will have $\mathbb{E}\left[\|h - \tilde{f}\|^2\right] \leq c_2^2 \varepsilon$ by Parseval's identity.

Note, however, that we may rescale $\varepsilon$ by constant factor $1/\left(2c_2^2\right)$ to obtain error $\varepsilon/2$ and only incur an additional constant (multiplicative) factor in the sample complexity bound. By the definition of $g$, we have

$$\|g\|^2 = \left(\sum_{S, \chi_S \notin C} \hat{f}(S)^2 + \sum_{S \in R} \hat{f}(S)^2\right) \tag{5}$$

where each $\hat{f}(S)$ for $S \in R$ is of magnitude at most $\varepsilon/4s$. By Fact 4 in Hazan et al. (2018) and Parseval's identity we have $\sum_R \hat{f}(R)^2 \leq \varepsilon/4$. Since $f$ is $(\varepsilon/4, s, C)$-concentrated we have $\sum_{S, \chi_S \notin C} \hat{f}(S)^2 \leq \varepsilon/4$. Thus, $\|g\|^2$ is at most $\varepsilon/2$. Therefore, by triangle inequality $\mathbb{E}\left[\|f - \tilde{f}\|^2\right] \leq \mathbb{E}\left[\|h - \tilde{f}\|^2\right] + \mathbb{E}\left[\|g\|^2\right] \leq \varepsilon$. It remains to bound $\|e\|_2$. Note that since the examples are chosen independently, the entries $e_i = g\left(z^i\right)$ are independent random variables. Since $g$ is a linear combination of orthonormal monomials (not including the constant term), we have $\mathbb{E}_{z \sim D}[g(z)] = 0$. Here we can apply linearity of variance (the covariance of $\chi_i$ and $\chi_j$ is zero for all $i \neq j$ ) and calculate the variance

$$\text{Var}\left(g\left(z^i\right)\right) = \left(\sum_{S, \chi_S \notin C} \hat{f}(S)^2 + \sum_{S \in R} \hat{f}(S)^2\right)$$

With the same calculation as (5), we know $\text{Var}\left(g\left(z^i\right)\right)$ is at most $\varepsilon/2$. Now consider the covariance matrix $V$ of the vector $e$ which equals $\mathbb{E}\left[ee^\top\right]$ (recall every entry of $e$ has mean 0). Then $V$ is a diagonal matrix (covariance between two independent samples is zero), and every diagonal entry is at

most $\varepsilon/2$. Applying Theorem 3 we have

$$\mathbb{P}\left(\|e\|_2 > \sqrt{\frac{\varepsilon}{2}}\delta\right) \leq \frac{m}{\delta^2}.$$

Setting $\delta = \sqrt{2m}$, we conclude that $\mathbb{P}\left(\|e\|_2 > \sqrt{\varepsilon m}\right) \leq \frac{1}{2}$. Hence with probability at least $1/2$, we have that $\|e\|_2 \leq \sqrt{\varepsilon m}$. From Theorem 5 in Hazan et al. (2018), we may choose $m = \tilde{O}\left(s^2/\varepsilon \cdot \log n^d\right)$. This completes the proof. Note that the probability $1/2$ above can be boosted to any constant probability with a constant factor loss in sample complexity.

For the running time complexity, we refer to Allen-Zhu & Yuan (2016) for optimizing linear regression with $\ell_1$ regularization. The running time is $O((T \log \frac{1}{\epsilon} + L/\epsilon) \cdot |C|)$, where $L$ is the smoothness of each summand in the objective. Since each $\chi_S$ takes value in $\{-1, 1\}$, the smoothness is bounded by the number of entries in each summand, which is $|C|$. Therefore, the running time is bounded by $O((T \log \frac{1}{\epsilon} + |C|/\epsilon) \cdot |C|)$. □

## C  LOW-DEGREE ALGORITHM

The low-degree algorithm is based on the concentration inequality, and it estimates the coefficient of each axis individually.

---
**Algorithm 2** Low-degree
---

1. Given uniformly randomly sampled $x_1, \cdots, x_T$, evaluate them on $f$: $\{f(x_1), ...., f(x_T)\}$.
2. For any $\chi_S \in C$, let $\hat{g}_S = \frac{\sum_{i=1}^T f(x_i)\chi_S(x_i)}{T}$.
3. Output the polynomial $g(x) = \sum_{S, \chi_S \in C} \hat{g}_S \chi_S(x)$.

---

**Theorem 4** (Linial et al. (1993)). *Given any $\epsilon, \delta > 0$, assuming that function $f$ is bounded by $B$, when $T \geq \frac{2B^2}{\epsilon^2} \log \frac{2|C|}{\delta}$, we have*

$$\Pr\left[\forall \chi_S \in C, s.t., |\hat{g}_S - \hat{f}_S| \leq \epsilon\right] \geq 1 - \delta$$

Theorem 4 was proved using the Hoeffding bound, and we included the proof here for completeness.

*Proof.* Since we are given $T$ samples to estimate $\hat{f}(S)$ for every $S$, we can directly apply the Hoeffding bound (notice that the function is bounded by $B$):

$$\Pr\left(|\alpha_S - \hat{f}(S)| \geq \epsilon\right) \cdot |C| \leq 2 \exp\left(-\frac{2T\epsilon^2}{4B^2}\right) = 2 \exp\left(-\frac{T\epsilon^2}{2B^2}\right)$$

Notice that $T \geq \frac{2B^2}{\epsilon^2} \log \frac{2|C|}{\delta}$, we know the right hand size is bounded by $\frac{\delta}{|C|}$, so Theorem 4 is proved. □

## D  DISCUSSION ON THE EXISTING ALGORITHMS

In this section, we compare our approach with the existing techniques from the perspectives of interpretation error and truthfulness.

**LIME (Ribeiro et al., 2016)**  Given an input $x$, Lime samples the neighborhood points based on a sampling distribution $\Pi_x$, and optimizes the following program:

$$\min_{g \in \mathcal{G}} L(f, g, \Pi_x) + \Omega(g)$$

where $L$ is the loss function describing the distance between $f$ and $g$ on the sampled data points, $\mathcal{G}$ is the set of readable functions (e.g. the set of linear functions), $\Omega(\cdot)$ is a function that characterizes the complexity of $g$. In other words, LIME tries to minimize the fitting error and simultaneously

minimizes the complexity of $g$ (which is usually the sparsity of the linear function). By minimizing $L$, LIME also works towards minimizing the interpretation error, but their approach is purely heuristic, without any theoretical guarantees. Although their readable function set can easily generalize to the set with higher order terms, the sampling distribution $\Pi_x$ is not uniform, so it is difficult to incorporate the orthonormal basis into their framework. In other words, the model they compute is not truthful.

**Attribution methods**   As we discussed in the introduction, attribution methods mainly focus on individual inputs, instead of the neighboring points. Therefore, it is difficult for the attribution methods to achieve low inconsistency, especially for first-order methods like SHAP (Lundberg & Lee, 2017) and IG (Sundararajan et al., 2017).

For higher-order attribution methods, consistency can potentially be improved due to their enhanced representation power. The classical Shapley interaction index has the problem of not precisely fitting the underlying function, as observed by Sundararajan et al. (2020), who proposed Shapley Taylor interaction index (Sundararajan et al., 2020) with better empirical performance. Shapley Taylor interaction index satisfies the generalized efficiency axiom, which says for all $f \in \{-1, 1\}^n \to \mathbb{R}$,

$$\sum_{S \subseteq [n], |S| \leq k} \mathcal{I}_S^k(f) = f([n]) - f(\emptyset)$$

We should remark that both the Shapley interaction index and Shapley Taylor interaction index were not originally designed for consistent interpretations, so they did not specify how to generalize the interpretation for the neighboring points. To this end, we make a global extension to the Shapley value based interpretation, that is, using Shapley interaction indices or Shapley Taylor interaction indices as the coefficients of corresponding terms of the polynomial surrogate function.

$$g(x_1, x_2, \cdots, x_n) = f(\emptyset) + \sum_{x_i \in S, S \subseteq [n]} \mathcal{I}(f, S)$$

However, these higher-order Shapley value based methods all focus on the original Shapley value framework, so their interpretations are not truthful, i.e., not getting the exact coefficient of $f$ even on the "simple bases". Moreover, as we will show in our experiments, higher-order methods still incur high interpretation errors compared with our methods.

When applying Shapley value techniques for visual search, Hamilton et al. (2022) proposed an interesting and novel sampling + Lasso regression algorithm for efficiently computing higher order Shapley Taylor index in their experiments. However, their methods are based on a sampling probability distribution generated from permutation numbers, which is far from the uniform distribution. Additionally, their algorithm is based on the Shapley Taylor index, so their method is not truthful as well.

## E   TEST WITH LOWER ORDER POLYNOMIAL FUNCTIONS

### E.1   FIRST ORDER POLYNOMIAL FUNCTION

To investigate the performance of different interpretation methods, let us take a closer look at a 1st order polynomial function:

$$f_1(x_1, x_2, x_3) := \frac{1}{2}x_1 - \frac{1}{3}x_2 + \frac{1}{4}x_3$$

For this simple function, we can manually compute the outcome of each algorithm, as illustrated in Table 1. If the algorithm's output is correct, i.e., equal to the output of $f_1$, we write a check mark. Otherwise, we write down the actual output of the given interpretation algorithm.

As we can see, all methods are consistent and efficient for all cases. In fact, all variants of Shapley indices degraded to 1st order Shapley values.

| Algorithms | $(-1,-1,-1)$ | $(-1,-1,+1)$ | $(-1,+1,-1)$ | $(-1,+1,+1)$ | $(+1,-1,-1)$ | $(+1,-1,+1)$ | $(+1,+1,-1)$ | $(+1,+1,+1)$ |
|---|---|---|---|---|---|---|---|---|
| **Ground Truth** | $-0.417$ | $+0.083$ | $-1.083$ | $-0.583$ | $+0.583$ | $+1.083$ | $-0.083$ | $+0.417$ |
| **LIME** | ✔ | ✔ | ✔ | ✔ | ✔ | ✔ | ✔ | ✔ |
| **SHAP** | ✔ | ✔ | ✔ | ✔ | ✔ | ✔ | ✔ | ✔ |
| **Shapley Interaction Index** (1st order) | ✔ | ✔ | ✔ | ✔ | ✔ | ✔ | ✔ | ✔ |
| **Shapley Taylor Index** (1st order) | ✔ | ✔ | ✔ | ✔ | ✔ | ✔ | ✔ | ✔ |
| **Faith-Shap** (1st order) | ✔ | ✔ | ✔ | ✔ | ✔ | ✔ | ✔ | ✔ |
| **Low-degree** (1st order) | ✔ | ✔ | ✔ | ✔ | ✔ | ✔ | ✔ | ✔ |
| **Harmonica** (1st order) | ✔ | ✔ | ✔ | ✔ | ✔ | ✔ | ✔ | ✔ |

Table 1: Interpretations by LIME, SHAP, Shapley Interaction Index, Shapley Taylor Index, Faith-Shap, Low-degree, and Harmonica on the 1st order polynomial function $f_1$.

### E.2 SECOND ORDER POLYNOMIAL FUNCTION

In addition, let us take a closer look at a 2nd order polynomial function:

$$f_2(x_1, x_2, x_3) := \frac{1}{2}x_1 - \frac{1}{3}x_2 + \frac{1}{4}x_3 - \frac{1}{5}x_1x_2 + \frac{1}{6}x_1x_3 - \frac{1}{7}x_2x_3$$

For this simple function, we can manually compute the outcome of each algorithm, as illustrated in Table 2. If the algorithm's output is correct, i.e., equal to the output of $f_2$, we write a check mark. Otherwise, we write down the actual output of the given interpretation algorithm.

As we can see, 2nd order interpretation algorithms, including Shapley Taylor index, Faithful Shapley, Low-degree, and Harmonica are consistent and efficient for all cases. Other methods can only fit a few inputs, the 2nd order Shapley interaction index misses all the cases because it is not efficient, and LIME misses all the cases because $f_2$ is not a linear function.

| Algorithms | $(-1,-1,-1)$ | $(-1,-1,+1)$ | $(-1,+1,-1)$ | $(-1,+1,+1)$ | $(+1,-1,-1)$ | $(+1,-1,+1)$ | $(+1,+1,-1)$ | $(+1,+1,+1)$ |
|---|---|---|---|---|---|---|---|---|
| **Ground Truth** | $-0.593$ | $-0.140$ | $-0.574$ | $-0.693$ | $+0.474$ | $+1.593$ | $-0.307$ | $+0.240$ |
| LIME | $-0.417$ | $+0.083$ | $-1.083$ | $-0.583$ | $+0.583$ | $+1.083$ | $-0.083$ | $+0.417$ |
| SHAP | $-0.240$ | $+0.283$ | $-1.250$ | $-0.726$ | $+0.726$ | $+1.250$ | $-0.283$ | ✔ |
| Shapley Interaction Index (2nd order) | $-0.329$ | $+0.171$ | $-0.995$ | $-0.781$ | $+0.671$ | $+1.505$ | $-0.395$ | $+0.152$ |
| **Shapley Taylor Index** (2nd order) | ✔ | ✔ | ✔ | ✔ | ✔ | ✔ | ✔ | ✔ |
| **Faithful Shapley** (2nd order) | ✔ | ✔ | ✔ | ✔ | ✔ | ✔ | ✔ | ✔ |
| **Low-degree** (2nd order) | ✔ | ✔ | ✔ | ✔ | ✔ | ✔ | ✔ | ✔ |
| **Harmonica** (2nd order) | ✔ | ✔ | ✔ | ✔ | ✔ | ✔ | ✔ | ✔ |

Table 2: Interpretations by LIME, SHAP, Shapley Interaction Index, Shapley Taylor Index, Faith-Shap, Low-degree, and Harmonica on the 2nd order polynomial function $f_2$.

### E.3 THIRD ORDER POLYNOMIAL FUNCTION

Finally, we investigate the following 3rd order polynomial, and present the result in Table 3.

$$f_3(x_1, x_2, x_3) := \frac{1}{2}x_1 - \frac{1}{3}x_2 + \frac{1}{4}x_3 - \frac{1}{5}x_1x_2 + \frac{1}{6}x_1x_3 - \frac{1}{7}x_2x_3 + \frac{1}{8}x_1x_2x_3$$

As we can see, Faith-Shap, Low-degree, and Harmonica are consistent and efficient for all cases. Other methods can only fit a few inputs, the 3rd order Shapley interaction index misses all the cases because it is not efficient, and LIME misses all the cases because $f_3$ is not a linear function.

| Algorithms | $(-1,-1,-1)$ | $(-1,-1,+1)$ | $(-1,+1,-1)$ | $(-1,+1,+1)$ | $(+1,-1,-1)$ | $(+1,-1,+1)$ | $(+1,+1,-1)$ | $(+1,+1,+1)$ |
|---|---|---|---|---|---|---|---|---|
| **Ground Truth** | $-0.718$ | $-0.015$ | $+0.449$ | $-0.818$ | $+0.599$ | $+1.468$ | $-0.432$ | $+0.365$ |
| LIME | $-0.417$ | $+0.083$ | $-1.083$ | $-0.583$ | $+0.583$ | $+1.083$ | $-0.083$ | $+0.417$ |
| SHAP | $-0.365$ | $+0.242$ | $-1.292$ | $-0.685$ | $+0.685$ | $+1.292$ | $-0.242$ | ✔ |
| Shapley Interaction Index (3rd order) | $-0.485$ | $+0.224$ | $-0.943$ | $-0.896$ | $+0.724$ | $+1.390$ | $-0.510$ | $+0.496$ |
| Shapley Taylor Index (3rd order) | ✔ | $+0.194$ | $-0.606$ | $-0.970$ | $+0.751$ | $+1.625$ | $-0.642$ | ✔ |
| **Faith-Shap** (3rd order) | ✔ | ✔ | ✔ | ✔ | ✔ | ✔ | ✔ | ✔ |
| **Low-degree** (3rd order) | ✔ | ✔ | ✔ | ✔ | ✔ | ✔ | ✔ | ✔ |
| **Harmonica** (3rd order) | ✔ | ✔ | ✔ | ✔ | ✔ | ✔ | ✔ | ✔ |

Table 3: Interpretations by LIME, SHAP, Shapley Interaction Index, Shapley Taylor Index, Faith-Shap, Low-degree, and Harmonica on the 3rd order polynomial function $f_3$.

## F   EXPERIMENT SETTINGS

In this section, we will describe the detailed experimental settings. For the two language tasks, i.e., SST-2 and IMDb, we use the same CNN neural network. The word embedding layer is pre-trained by GloVe (Pennington et al., 2014) and the maximum word number is set to 25, 000. Besides the embedding layer, the network consists of several convolutional kernels with different kernel sizes (3, 4, and 5). After that, we use several fully connected layers, non-linear layers, and pooling layers to process the features. A Sigmoid function is attached to the tail of the network to ensure that the output can be seen as a probability distribution. Our networks are trained with an Adam (Kingma & Ba, 2015) optimizer with a learning rate of 0.01 for 5 epochs. For the vision task, we choose the official ResNet (He et al., 2016) architecture, which is available on PyTorch and we do not discuss the architecture details here.

Notice that our algorithm is a post-hoc model-agnostic interpretation algorithm, we only need to use the original neural network $f$ to infer on given input as an oracle. This means that one can easily change the network architecture without any additional changes.

All the experiments are run on a server with 4 Nvidia 2080 Ti GPUs. More information about the runtime python environment and implementation details can be found in our code.

## G   DETAILED NUMERICAL RESULTS

In this section, we provide numerical results for Figure 2, 4, and 6 in Table 4, 5 and 6, respectively.

| radius | $L^2$ norm | $L^1$ norm | $L^0$ norm |
|--------|-----------|-----------|-----------|
| 1 | 0.0536 | 0.0460 | 0.1154 |
| 2 | 0.0576 | 0.0463 | 0.1219 |
| 4 | 0.0639 | 0.0483 | 0.1350 |
| 8 | 0.0768 | 0.0549 | 0.1696 |
| 16 | 0.0945 | 0.0651 | 0.2173 |
| 32 | 0.0990 | 0.0677 | 0.2275 |
| $\infty$ | 0.0991 | 0.0677 | 0.2279 |

**Harmonica$^2$**

| radius | $L^2$ norm | $L^1$ norm | $L^0$ norm |
|--------|-----------|-----------|-----------|
| 1 | **0.0434** | 0.0363 | 0.0995 |
| 2 | **0.0484** | 0.0376 | 0.1046 |
| 4 | **0.0561** | 0.0405 | 0.1157 |
| 8 | **0.0700** | 0.0474 | 0.1456 |
| 16 | **0.0876** | 0.0575 | 0.1900 |
| 32 | **0.0921** | 0.0600 | 0.2003 |
| $\infty$ | **0.0922** | 0.0601 | 0.2005 |

**Harmonica$^3$**

| radius | $L^2$ norm | $L^1$ norm | $L^0$ norm |
|--------|-----------|-----------|-----------|
| 1 | 0.0981 | 0.0870 | 0.3572 |
| 2 | 0.1010 | 0.0861 | 0.3458 |
| 4 | 0.1043 | 0.0863 | 0.3421 |
| 8 | 0.1135 | 0.0919 | 0.3659 |
| 16 | 0.1285 | 0.1015 | 0.4034 |
| 32 | 0.1323 | 0.1038 | 0.4111 |
| $\infty$ | 0.1322 | 0.1037 | 0.4112 |

LIME

| radius | $L^2$ norm | $L^1$ norm | $L^0$ norm |
|--------|-----------|-----------|-----------|
| 1 | 0.0792 | 0.0568 | 0.1819 |
| 2 | 0.1089 | 0.0836 | 0.3167 |
| 4 | 0.1470 | 0.1186 | 0.4847 |
| 8 | 0.1865 | 0.1554 | 0.6252 |
| 16 | 0.2106 | 0.1783 | 0.6891 |
| 32 | 0.2137 | 0.1814 | 0.6958 |
| $\infty$ | 0.2137 | 0.1814 | 0.6961 |

SHAP

| radius | $L^2$ norm | $L^1$ norm | $L^0$ norm |
|--------|-----------|-----------|-----------|
| 1 | 0.1133 | 0.0681 | 0.2010 |
| 2 | 0.1551 | 0.1055 | 0.3361 |
| 4 | 0.2081 | 0.1563 | 0.5066 |
| 8 | 0.2595 | 0.2075 | 0.6476 |
| 16 | 0.2852 | 0.2336 | 0.7058 |
| 32 | 0.2872 | 0.2357 | 0.7099 |
| $\infty$ | 0.2874 | 0.2359 | 0.7105 |

Integrated Gradients

| radius | $L^2$ norm | $L^1$ norm | $L^0$ norm |
|--------|-----------|-----------|-----------|
| 1 | 0.0865 | 0.0477 | 0.1215 |
| 2 | 0.1232 | 0.0756 | 0.2118 |
| 4 | 0.1753 | 0.1183 | 0.3406 |
| 8 | 0.2343 | 0.1691 | 0.4859 |
| 16 | 0.2678 | 0.1994 | 0.5724 |
| 32 | 0.2709 | 0.2028 | 0.5823 |
| $\infty$ | 0.2711 | 0.2029 | 0.5823 |

Integrated Hessians

| radius | $L^2$ norm | $L^1$ norm | $L^0$ norm |
|--------|-----------|-----------|-----------|
| 1 | 0.0623 | 0.0472 | 0.1090 |
| 2 | 0.0853 | 0.0649 | 0.1971 |
| 4 | 0.1144 | 0.0874 | 0.3069 |
| 8 | 0.1426 | 0.1081 | 0.3909 |
| 16 | 0.1612 | 0.1217 | 0.4404 |
| 32 | 0.1642 | 0.1241 | 0.4486 |
| $\infty$ | 0.1642 | 0.1240 | 0.4489 |

Shapley Taylor Index$^2$

| radius | $L^2$ norm | $L^1$ norm | $L^0$ norm |
|--------|-----------|-----------|-----------|
| 1 | 0.0602 | 0.0451 | 0.1034 |
| 2 | 0.0813 | 0.0614 | 0.1795 |
| 4 | 0.1091 | 0.0824 | 0.2794 |
| 8 | 0.1387 | 0.1042 | 0.3707 |
| 16 | 0.1586 | 0.1190 | 0.4271 |
| 32 | 0.1616 | 0.1213 | 0.4356 |
| $\infty$ | 0.1616 | 0.1213 | 0.4357 |

Faith-Shap$^2$

Table 4: The interpretation error of Harmonica and other baseline algorithms evaluated on the SST-2 dataset for different neighborhoods with a radius ranging from 1 to $\infty$ under $L^2$, $L^1$ and $L^0$ norm.

## H   DISCUSSION ON THE LOW-DEGREE ALGORITHM

From Theorem 2 and Theorem 4, we know that the sample complexity of the Harmonica algorithm ($\tilde{O}(\frac{1}{\epsilon})$) is much more efficient than the Low-degree algorithm ($\tilde{O}(\frac{1}{\epsilon^2})$). Figure 8 shows that when evaluating the interpretation error on SST-2 dataset, with the same sample size, the Harmonica

| radius | $L^2$ norm | $L^1$ norm | $L^0$ norm |
|---|---|---|---|
| 1 | 0.0224 | 0.0192 | 0.0040 |
| 2 | 0.0234 | 0.0193 | 0.0035 |
| 4 | 0.0300 | 0.0228 | 0.0140 |
| 8 | 0.0441 | 0.0298 | 0.0433 |
| 16 | 0.0506 | 0.0329 | 0.0573 |
| 32 | 0.0514 | 0.0334 | 0.0595 |
| $\infty$ | 0.0514 | 0.0334 | 0.0596 |

| radius | $L^2$ norm | $L^1$ norm | $L^0$ norm |
|---|---|---|---|
| 1 | **0.0175** | 0.0149 | 0.0027 |
| 2 | **0.0192** | 0.0157 | 0.0022 |
| 4 | **0.0264** | 0.0195 | 0.0098 |
| 8 | **0.0407** | 0.0265 | 0.0367 |
| 16 | **0.0472** | 0.0296 | 0.0505 |
| 32 | **0.0481** | 0.0301 | 0.0527 |
| $\infty$ | **0.0481** | 0.0301 | 0.0528 |

| radius | $L^2$ norm | $L^1$ norm | $L^0$ norm |
|---|---|---|---|
| 1 | 0.0624 | 0.0525 | 0.1175 |
| 2 | 0.0671 | 0.0546 | 0.1358 |
| 4 | 0.0740 | 0.0583 | 0.1606 |
| 8 | 0.084 | 0.0640 | 0.1903 |
| 16 | 0.0885 | 0.0664 | 0.2017 |
| 32 | 0.0892 | 0.0667 | 0.2033 |
| $\infty$ | 0.0892 | 0.0667 | 0.2033 |

**Harmonica$^2$**    **Harmonica$^3$**    LIME

| radius | $L^2$ norm | $L^1$ norm | $L^0$ norm |
|---|---|---|---|
| 1 | 0.0849 | 0.0650 | 0.2266 |
| 2 | 0.1123 | 0.0903 | 0.3589 |
| 4 | 0.1406 | 0.1162 | 0.4891 |
| 8 | 0.1598 | 0.1341 | 0.5610 |
| 16 | 0.1662 | 0.1402 | 0.5817 |
| 32 | 0.1671 | 0.1411 | 0.5842 |
| $\infty$ | 0.1671 | 0.1411 | 0.5843 |

| radius | $L^2$ norm | $L^1$ norm | $L^0$ norm |
|---|---|---|---|
| 1 | 0.1228 | 0.0810 | 0.2702 |
| 2 | 0.1576 | 0.1169 | 0.4102 |
| 4 | 0.1882 | 0.1494 | 0.5333 |
| 8 | 0.2031 | 0.1657 | 0.5885 |
| 16 | 0.2067 | 0.1697 | 0.6004 |
| 32 | 0.2071 | 0.1702 | 0.6014 |
| $\infty$ | 0.2071 | 0.1702 | 0.6015 |

| radius | $L^2$ norm | $L^1$ norm | $L^0$ norm |
|---|---|---|---|
| 1 | 0.1178 | 0.0635 | 0.1929 |
| 2 | 0.1567 | 0.0997 | 0.3125 |
| 4 | 0.1969 | 0.1415 | 0.4501 |
| 8 | 0.2227 | 0.1702 | 0.5423 |
| 16 | 0.2308 | 0.1795 | 0.5702 |
| 32 | 0.2319 | 0.1807 | 0.5739 |
| $\infty$ | 0.2319 | 0.1807 | 0.5738 |

SHAP    Integrated Gradients    Integrated Hessians

| radius | $L^2$ norm | $L^1$ norm | $L^0$ norm |
|---|---|---|---|
| 1 | 0.0444 | 0.0365 | 0.0554 |
| 2 | 0.0590 | 0.0470 | 0.0999 |
| 4 | 0.0684 | 0.0538 | 0.1339 |
| 8 | 0.0698 | 0.0542 | 0.1300 |
| 16 | 0.0719 | 0.0558 | 0.1287 |
| 32 | 0.0741 | 0.0575 | 0.1294 |
| $\infty$ | 0.0743 | 0.0577 | 0.1294 |

| radius | $L^2$ norm | $L^1$ norm | $L^0$ norm |
|---|---|---|---|
| 1 | 0.0395 | 0.0313 | 0.0337 |
| 2 | 0.0469 | 0.0378 | 0.0466 |
| 4 | 0.0547 | 0.0440 | 0.0657 |
| 8 | 0.0620 | 0.0494 | 0.0808 |
| 16 | 0.0684 | 0.0543 | 0.0851 |
| 32 | 0.0740 | 0.0588 | 0.0859 |
| $\infty$ | 0.0744 | 0.0590 | 0.0859 |

Shapley Taylor Index$^2$    Faith-Shap$^2$

Table 5: The interpretation error of Harmonica and other baseline algorithms evaluated on the IMDb dataset for different neighborhoods with a radius ranging from 1 to $\infty$ under $L^2$, $L^1$ and $L^0$ norm.

| radius | $L^2$ norm | $L^1$ norm | $L^0$ norm |
|---|---|---|---|
| 1 | 0.1290 | 0.1108 | 0.4248 |
| 2 | 0.1308 | 0.1073 | 0.4116 |
| 4 | 0.1373 | 0.1094 | 0.4332 |
| 8 | 0.1584 | 0.1264 | 0.5156 |
| $\infty$ | 0.1693 | 0.1342 | 0.5405 |

| radius | $L^2$ norm | $L^1$ norm | $L^0$ norm |
|---|---|---|---|
| 1 | **0.1048** | 0.0880 | 0.3202 |
| 2 | **0.1108** | 0.0887 | 0.3260 |
| 4 | **0.1220** | 0.0955 | 0.3723 |
| 8 | **0.1443** | 0.1139 | 0.4698 |
| $\infty$ | **0.1566** | 0.1230 | 0.5030 |

| radius | $L^2$ norm | $L^1$ norm | $L^0$ norm |
|---|---|---|---|
| 1 | 0.2422 | 0.2208 | 0.7274 |
| 2 | 0.2347 | 0.2036 | 0.6976 |
| 4 | 0.2346 | 0.1918 | 0.6540 |
| 8 | 0.2897 | 0.2304 | 0.6893 |
| $\infty$ | 0.3261 | 0.2579 | 0.7196 |

**Harmonica$^2$**    **Harmonica$^3$**    LIME

| radius | $L^2$ norm | $L^1$ norm | $L^0$ norm |
|---|---|---|---|
| 1 | 0.1197 | 0.0867 | 0.2892 |
| 2 | 0.1658 | 0.1261 | 0.4566 |
| 4 | 0.2306 | 0.1862 | 0.6483 |
| 8 | 0.2943 | 0.2409 | 0.7318 |
| $\infty$ | 0.3115 | 0.2523 | 0.7362 |

| radius | $L^2$ norm | $L^1$ norm | $L^0$ norm |
|---|---|---|---|
| 1 | 0.2322 | 0.1650 | 0.4871 |
| 2 | 0.3141 | 0.2375 | 0.6406 |
| 4 | 0.4200 | 0.3346 | 0.7722 |
| 8 | 0.5010 | 0.4094 | 0.8300 |
| $\infty$ | 0.5113 | 0.4177 | 0.8340 |

| radius | $L^2$ norm | $L^1$ norm | $L^0$ norm |
|---|---|---|---|
| 1 | 1.3681 | 0.9504 | 0.8443 |
| 2 | 1.8603 | 1.4006 | 0.9025 |
| 4 | 2.5168 | 2.0427 | 0.9394 |
| 8 | 3.1095 | 2.6844 | 0.9543 |
| $\infty$ | 3.2139 | 2.8140 | 0.9562 |

SHAP    Integrated Gradients    Integrated Hessians

| radius | $L^2$ norm | $L^1$ norm | $L^0$ norm |
|---|---|---|---|
| 1 | 0.1337 | 0.1039 | 0.3939 |
| 2 | 0.1681 | 0.1309 | 0.4906 |
| 4 | 0.2028 | 0.1600 | 0.5830 |
| 8 | 0.2226 | 0.1771 | 0.6365 |
| $\infty$ | 0.2274 | 0.1804 | 0.6418 |

| radius | $L^2$ norm | $L^1$ norm | $L^0$ norm |
|---|---|---|---|
| 1 | 0.1238 | 0.0948 | 0.3443 |
| 2 | 0.1499 | 0.1161 | 0.4338 |
| 4 | 0.1718 | 0.1351 | 0.5126 |
| 8 | 0.1960 | 0.1554 | 0.5872 |
| $\infty$ | 0.2099 | 0.1654 | 0.6071 |

Shapley Taylor Index$^2$    Faith-Shap$^2$

Table 6: The interpretation error of Harmonica and other baseline algorithms evaluated on the ImageNet dataset for different neighborhoods with a radius ranging from 1 to $\infty$ under $L^2$, $L^1$ and $L^0$ norm.

algorithm outperforms the Low-degree algorithm by a large margin. We further increase the sample size for the Low-degree algorithm and see that its interpretation error gradually approaches that of Harmonica. However, even with 5x sample size, the Low-degree algorithm still gives a larger interpretation error compared with Harmonica.

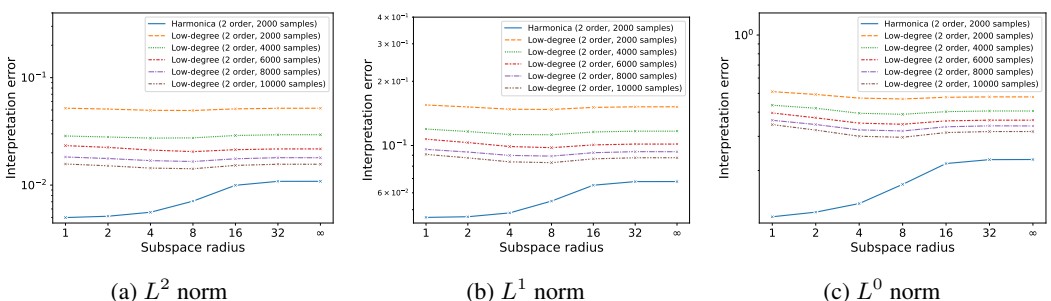

(a) $L^2$ norm  (b) $L^1$ norm  (c) $L^0$ norm

Figure 8: Visualization of interpretation error $\mathbb{I}_{p, \mathcal{N}_x}(f, g)$ evaluated on SST-2 dataset, while Harmonica and Low-degree algorithms using different sample size varying from 2000 to 10000.

