# OpenReview forum: "Consistent and Truthful Interpretation with Fourier Analysis"
_ICLR.cc/2023/Conference — Submitted to ICLR 2023_

### Official Review · Reviewer_BSNn · 2022-10-23

**Confidence:** 4
**Correctness:** 2
**Technical Novelty And Significance:** 2
**Empirical Novelty And Significance:** 2
**Recommendation:** 5

**Clarity, Quality, Novelty And Reproducibility:**

Presentation quality should be improved. For reproducibility, experimental details need further description.

**Strength And Weaknesses:**


Strong Points:
[1]This paper produced a novel method of Fourier analysis of Boolean function to get consistency guarantees.


Weak Points:
[1]It is suggested that the author need to highlight the contributions of this paper, especially the part of technique improvements.
[2]This paper lacks an understanding of where the paper's going, the authors, for example, fail to focus on the consistency of interpretability and the improvements of truth interpretation.
[3]As above, it needs to review the works of about ML inconsistent rather than the different types of interpretable models in Related Work.
[4]For any model  and  especially with different structure, the weight distribution which is defined to evaluate the inconsistency by Equation (1) is lack of some evidence. It might be needed to provide some demonstrations.
[5]It is better here to supplement more clear and plain explanations about algorithm 1 and 2.
[6]How to evaluate that a function can faithfully interprets the readable part of the network or not in section 3.2?
[7]Although the experiment results based on SST-2 and IMDb datasets is impressive compared with other baselines, it is needed more clear and sufficient analysis about that. And the experimental hardware environment is not reported.
[8]Many of the reference should be further improved. For example, few references miss page, and the format of references is not uniform and either AAAI or its full name. In addition, the authors may try to discuss the existing work in published papers (rather than a number of preprinted references from arxiv).

**Summary Of The Paper:**

Focus on the consistent of ML interpretations, this paper introduces a new objective of consistency based on a notion called truthful interpretation by applying Fourier analysis of Boolean functions. Experimental results show that the method achieves higher consistency compared with other methods.

**Summary Of The Review:**

Contributions, especailly technique improvement need to be cleary addressed.

---

> ### Author Response · Authors · 2022-11-08
> **To Reviewer BSNn**
>
> Thank you for your time reviewing our paper and your valuable feedback!
>
> As we mentioned in the general response, our main contribution is not proposing Harmonica as another interpretation algorithm. Instead, we have made three other contributions (see general response), and now they are explicitly stated in Introduction.
>
> Answer to your questions:
>
> ---
>
> **Q: This paper lacks an understanding of where the paper’s going.**
>
> A: Thank you for pointing it out! The previous presentation was indeed confusing, and we spent too much space explaining Fourier analysis. Now we have completely moved Fourier analysis and algorithm descriptions, theoretical guarantees to the Appendix (as they are not new). Moreover, we have rewritten the introduction, adding three questions with bounding box, to clearly convey the main theoretical contribution of our paper. Hopefully this new version has better presentation of our results.
>
> ---
> **Q: Related work on inconsistent algorithms?**
>
> A: We did not include this part previously, because the consistency property, despite being very important for many applications, was somewhat overlooked in the past. Efficiency was a much more popular property. There were indeed a few algorithms that are consistent (LIME as the most famous example), but they are mainly heuristic driven approaches. We have added a paragraph discussing these algorithms in the related work.
>
> ---
>
> **Q: Definition of inconsistency is weird, why choose $\ell_2^2$ norm?**
>
> A: That is indeed a very good point. We have modified the definition (renamed to interpretation error) to be the $\ell_p$ distance in the function space. More specifically, when we have a target network $f$, a interpreting network $g$, the interpretation error is the $\ell_p$ distance between $f$ and $g$, under any measure $\mu$. This new definition is more natural and general, and thank you for your suggestion!
>
> ---
> **Q: More explanations about Algorithm 1 and 2?**
>
> A: Certainly. The new introduction and framework of interpretability should be much easier to understand, and we also added a few sentences explaining the algorithms in Section 4. Intuitively, both algorithms are just learning the weights/coefficients of different feature interactions, like $x_1$, $x_2x_3$, $x_2x_6x_7$, etc. One can show that these two algorithms can accurately learn these coefficients, without any biases.
>
> ---
>
> ***Q: How to evaluate that a function can faithfully interprets the readable part of the network or not?***
>
> A: We apologize for the confusion. The readable part, by our definition, is essentially a subspace in the function space. If a model can faithfully interpret the readable part, it means it equals the target network in this subspace. In other words, restricted to this subspace, the model and network have the same representation, or equivalently, have the same coefficients for the bases supporting this subspace. We have added Lemma 1 describing this derivation in details, and also discussed how to estimate the truthful gap by sampling in Section 5.3.
>
> ---
>
> ***Q: Experimental results should be improved, and hardware environment should be clearly described.***
>
> A: Sure. We have significantly updated the experimental results, and also added the hardware environment. Please check our general response for the details.
>
> ---
>
> ***Q: Reference should be improved.***
>
> A: Thank you for the suggestion! We have made the changes accordingly.
>
> ---
>
> We are very grateful for your suggestions on improving our paper! Following your suggestion, we have made significant changes in the revision, and we hope you can take a look!

---

### Official Review · Reviewer_hMRo · 2022-10-23

**Confidence:** 5
**Correctness:** 2
**Technical Novelty And Significance:** 2
**Empirical Novelty And Significance:** 2
**Recommendation:** 3

**Clarity, Quality, Novelty And Reproducibility:**

The idea of this paper is clear and the paper is well written. However, the theoretical novelty is limited and experimental settings are unclear.

**Strength And Weaknesses:**

[Strengths]
1. This paper focused on an important topic, i.e., learning consistent and truthful explanations.
2. This paper is well written. It is easy to follow the authors’ ideas.

[Weaknesses]

1. There exists a significant theoretical defect in the algorithm. Algorithm 1 and Algorithm 2 aim to learn the polynomial $g(x)=\sum_{S,\chi_S\in C} \alpha_S \chi_S(x)$ as the explanation model. However, such an explanation model cannot provide universal explanations because it does not consider the dimension alignment issue. For example, given two sentences, “How are you.” and “Hello, how are you.”, $\chi_{\\{x_1, x_2, x_3\\}}$ and $\chi_{\\{x_2, x_3, x_4\\}}$ both correspond to the sub-sentence “how are you” of these two sentences. Therefore, the network is supposed to predict the same score for the input of these two sub-sentences, but the explanation model provides different explanations, i.e., $\alpha_1\chi_{\\{x_1, x_2, x_3\\}}$ and $\alpha_2\chi_{\\{x_2, x_3, x_4\\}}$. Such explanations do not make any sense. It seems that such explanation models are only suitable for data where each dimension has a fixed meaning. This is our main concern.

2. The authors do not provide explanation results of the proposed method. Therefore, it is difficult for us to make any comment on the interpretability of the proposed method.

3. The metric of consistency is unable to strictly measure the consistency between normal samples and masked samples. It is because the masked sample is an out-of-distribution (OOD) sample. The explainable model is supposed to provide different explanations for normal samples and OOD samples. For example, a normal sentence contains ten words, and the masked sentence only contains two words, whose explanations are almost absolutely inconsistent. In this way, even the supposed optimal explanation is unlikely to get the inconsistency of zero, i.e., $I_D(f,g)\ne 0$. The authors are supposed to prove that your definition of inconsistency can correctly reflect the absolute inconsistency of the explanation method.

4. The authors’ three claims about “interpretability, consistency, and efficiency” lack clear definition and proof, i.e., “(a) Attribution methods are interpretable and efficient, but not consistent. (b) The original network is consistent and efficient but not interpretable. (c) If one model is interpretable and consistent, it cannot be efficient.” I find that it is very difficult for me to provide any comments on these unprecise claims. The authors are supposed to provide a clear definition of “interpretability, consistency, and efficiency,” and provide proofs of these three claims.

5. Concern about the correctness of the input space of functions $f$ and $g$. In some Definitions, e.g., Definition 3.2 and Definition 3.3, $f,g\in\\{-1,1\\}^n$, but in other Definitions, e.g., Definition 3.4 and Definition 3.6, $f,g\in\\{0,1\\}^n$. If they are all correct, then the authors are supposed to explain the reason why the input space is different.

6. Definition 3.8 gives the lower bound of the fitting error of the explanation model. Why do you focus on the lower bound of the fitting error, instead of the upper bound?

7. The experimental comparisons of the proposed explanation method and other explanation methods in Section 5.1 are unfair. It is because the majority function contains the product relation $x_1x_2x_3$, which naturally prefers the explanation method proposed in this paper, where $\chi_S$ is also defined on the product relation $\prod_{i\in S} x_i$. The authors are supposed to conduct experimental comparisons on other functions that do not depend on the product relation of input variables.

8. Ask for more experiments. (1) The authors are supposed to compare the inconsistency between the proposed method and the Faith-SHAP [cite1], which is also an efficient explanation method. (2) The authors are supposed to conduct experimental comparisons with more models, such as LSTMs or Transformers. (3) The authors are supposed to conduct experiments to verify the effectiveness of Algorithm 2.

[cite1] Tsai et al. “FAITH-SHAP: THE FAITHFUL SHAPLEY INTERACTION INDEX” in arXiv:2203.00870

9. In Figure 3, results show that the proposed method did not exhibit significantly lower readable inconsistency than other methods. This makes me question the effectiveness of the proposed method.

Minor.
- Which convolutional neural network is the authors use for experiments in Section 6.1?

- Please provide the actual radius in Figures 2, 3, and 4, instead of $\infty$

- In Definition 3.1, I think it should be $\chi_S(x)$ instead of $\chi_S$ in the left-hand side of the equation.


**Summary Of The Paper:**

This paper proposes an explanation method to provide consistent and truthful explanations of black-box models. Experimental results verify the effectiveness of the proposed method.

**Summary Of The Review:**

There exists a significant theoretical defect of the proposed method and the authors do not provide explanation results of the proposed method. Besides, experimental settings are unclear.

---

> ### Author Response · Authors · 2022-11-08
> **To Reviewer hMRo**
>
> Thank you for the in-depth comments. Many of them are very insightful, which greatly improved our paper!
>
> Answer to your questions:
>
> ---
>
> **Q: [How are you] vs [Hello, how are you]. Your algorithm fails in this case?**
>
> A: Right, that is a VERY good point. The short answer is, our algorithm is under the "remove-based explanation" category (Covert et al 2021), which does not consider the case that the input space will change. To the best of our knowledge, all the existing interpretable methods (including LIME, SHAP, Shapley-Taylor and Faith-SHAP) also belong to this category (26 of them were discussed in Covert et al 2021). For many scenarios, e.g., healthcare, remove-based explanations are the right framework with enough power. Therefore, this is not a theoretical defect of our paper (or, it is the theoretical defect of all the existing interpretable models).
>
> But we understand that this question is mainly because we did not formally define consistency in our original draft. In our revision, we have formally defined it in Definition 3, and also add a discussion at the end of Section 3.
>
> ---
>
> **Q: The explanation results were not provided.**
>
> A: We have added the manually computed results of lower order polynomials in Appendix E. Hopefully these examples will provide some intuitions of our algorithms.
>
> ---
> **Q: The metric of consistency is not a good metric, due to the OOD samples.**
>
> A: That’s another very good point! We have modified the definition (renamed to interpretation error) to be the $\ell_p$ distance in the function space. More specifically, when we have a target network $f$, an interpreting network $g$, the interpretation error is the $\ell_p$ distance between $f$ and $g$, under any measure $\mu$. This new definition is more natural and general, and thank you for your suggestion!
>
> ---
> **Q: The claim of impossible trinity is not precise.**
>
> A: You are right. In our revision, we have added a theorem with rigorous definitions for the impossible trinity. Thank you for the suggestion!
>
> ---
>
> **Q: Input domain is {0,1}^n or {-1,1}^n?**
>
> A: Those were typos. We meant {-1,1}^n throughout the paper, and sorry for the confusion.
>
> ---
>
> **Q: Why does Definition 3.8 give lower bound?**
>
> A: You are right, it’s a typo. We’ve modified the definition and moved it to Appendix B.
>
> ---
>
> **Q: Unfair comparison in the polynomial setting?**
>
> A: You are right. We have added two more cases, i.e., 1st order and 2nd order polynomials, to give more comprehensive discussion on different algorithms in Appendix E.
>
> ---
>
> **Q: You should compare with Faith-SHAP.**
>
> A: Thank you for your suggestion! Faith-SHAP is a very insightful paper with delicate theory. We have compared it in the polynomial setting, and found that in theory it is able to get all the coefficients accurately, just like our algorithms, and is much better than other higher order methods.
> Faith-SHAP has a delicate representation theorem, which assigns coefficients to different terms under the Mobius transform. Since the basis induced by the Mobius transform is not orthonormal, it is not clear to us whether Faith-SHAP can theoretically compute the accurate coefficients for higher order functions. However, the running time of Faith-SHAP has exponential dependency on $n$, so empirically weighted sampling on subsets of features is needed Tsai et al 2022. This might be the main reason that our algorithms outperform Faith-SHAP in the experiments with real datasets.
>
> ---
>
> **Q: Ask for more experiments.**
>
> A:  We have significantly updated the experimental results. Please check our general response for the details.
>
> ---
>
> **Q: Figure 3 is confusing.**
>
> A: Right, we have modified the presentation in our revision in Figure 3,5,7.
>
> ---
>
> **Q: What is the convolutional network structure?**
>
> A: We have uploaded the code as the supplementary materials. For the two language tasks, i.e., SST-2 and IMDb, we use the same CNN neural network. The word embedding layer is pre-trained by GloVe and the maximum word number is set to 25000. Besides the embedding layer, the network consists of several convolutional kernels with different kernel sizes (3, 4, and 5). After that, we use several fully connected layers, non-linear layers, and pooling layers to process the features. A Sigmoid function is attached to the tail of the network to ensure that the output can be seen as a probability distribution. Our networks are trained with an Adam optimizer with a learning rate of 0.01 for 5 epochs. For the vision task, we choose the official ResNet architecture, which is available on PyTorch.
>
> ---
>
> **Q: What is the actual radius of Figure 2,3,4? Please do not use $\infty$.**
>
> A: We use $\infty$ to represent the maximum sentence length, which may vary for different data points, so we cannot use a fixed number on the figure.
>
> ---
>
> Again, thank you for your detailed and thoughtful comments! Those were very helpful :)

---

### Official Review · Reviewer_dqjk · 2022-10-25

**Confidence:** 3
**Correctness:** 4
**Technical Novelty And Significance:** 2
**Empirical Novelty And Significance:** 2
**Recommendation:** 3

**Clarity, Quality, Novelty And Reproducibility:**

The paper is fairly well-written.

The goal of the paper seems to be to examine the tension between consistency and simplicity in the context of interpretability. However, the insights provided by this paper are limited to Boolean functions where the set of interpretable functions is taken to be functions that are sparse in Fourier basis. As a result, it is not clear whether the insights are applicable to practical settings. Moreover, the algorithmic and empirical contribution of the paper are both limited.



**Strength And Weaknesses:**

Strengths
- The paper illustrates that existing tools from learning sparse approximations to Boolean functions could be related to interpretability.

Weaknesses
- The analysis in the paper is entirely restricted to Boolean functions, and the restricted function class is assumed to be sparse functions in the Fourier basis. The algorithms studied in this paper rely heavily on these assumptions.
- The algorithms investigated by the paper are standard in the learning Boolean functions literature (as the paper does acknowledge). For example, Section 4 is entirely devoted to analyzing existing algorithms (with minor modifications).
- The empirical evaluation is restricted to small-scale setups: a convolutional network achieving ~80% accuracy on the SST-2 dataset and a convolutional network achieving trained on the IMDb dataset. It would be useful to more extensively empirical evaluate their methods on larger-scale setups with modern networks.

**Summary Of The Paper:**

The paper investigates the tension between consistency and simplicity in interpretability. The goal is to find a model from a restricted function class (e.g. of simple functions) that matches the network on as much of the input space as possible. The paper focuses on Boolean functions, takes the function space to be functions that are space in the Fourier basis, and investigates approaches to find the sparse function that best approximates a given function. They investigate the performance of (slightly modified versions of) the Harmonica algorithm (Hazan et al., 2017) and Low-degree (Linial et al., 1993). The paper then evaluates these algorithms and compares them to existing approaches—including LIME and SHAP—on datasets for sentiment analysis and movie review classification.

**Summary Of The Review:**

The paper investigates the tensions between consistency and simplicity in the context of interpretability, with a focus on interpreting Boolean functions with functions that are sparse in Fourier basis: however, the setup is quite restrictive, and furthermore, both the algorithmic and empirical contributions are limited.

---

> ### Author Response · Authors · 2022-11-08
> **To Reviewer dqjk**
>
> Thank you for your time reviewing our paper and for your valuable comments!
>
> As we mentioned in the general response, our main contribution is not proposing Harmonica as another interpretation algorithm. Instead, we have made three other contributions (see general response), and now they are explicitly stated in Introduction.
>
> Answer to your questions:
>
> ------------------------------------------------
> **Q: Boolean function seems to be very restricted? And you need the sparsity assumption?**
>
> A: Boolean function is much more powerful than it seems to be. In Boolean functions, only the input variables are Boolean, and the outputs are real numbers. This is the commonly used assumption in the literature. For example, LIME, SHAP, Shapley Taylor, and many other methods are all (implicitly or explicitly) using Boolean functions. Boolean functions are ideal for generating interpretations, e.g., how does the existence (Yes/No) of factor A affect the outcome?
>
> Moreover, Boolean functions have very strong representation power. For example, empirically most human readable interpretations can be converted into (ensemble of) decision trees, and theoretically all (ensemble of) decision trees can be converted into Boolean functions. The widely used algorithm XGBoost is based on an ensemble of decision trees.
>
> Our algorithm Harmonica needs the assumption that the Boolean function is sparse, but Low-degree does not need this. Therefore, as we mentioned in our paper, for functions not sparse, Low-degree can deal with them theoretically. Moreover, our main contribution is not proposing these two existing algorithms; instead, we identified that learning Boolean functions is the unique solution for finding consistent and truthful interpretations for Boolean functions. Therefore, the sparsity assumption does not have strong impact on our contribution.
>
> ------------------------------------------------
> **Q: The algorithm and analysis in this paper looks standard and incremental to the existing work?**
>
> A: Thank you for pointing it out! You are right, that is a very bad presentation of our paper. Now we have completely moved these parts into appendix, and conveyed our contributions in the main paper.
>
> ------------------------------------------------
> **Q: Maybe you should use large-scale modern networks?**
>
> A: Yes, you are right. It makes sense to try our algorithm on large-scale modern networks. Like LIME and SHAP, our algorithm is a post-hoc interpretation algorithm, which means our algorithm only relies on the inference stage of neural networks, not the training stage. In other words, using larger network is straightforward for our algorithm. We have added an experiment on ImageNet with Resnet-101 (with 44,549,160 parameters), and got equally good experimental results.
>
> ------------------------------------------------
> **Q: Can your algorithms be used in the practical settings?**
>
> A: Yes, our algorithm can be easily applied to the practical settings, just like LIME and SHAP, in the post-hoc manner. In terms of running time, our algorithm is equally fast as LIME, and is much faster than SHAP, because SHAP needs to apply sampling among exponentially many subsets. We will release a python package of our algorithm pretty soon.
>
> ------------------------------------------------
>
> Again, we are sorry for the misleading presentation of our original version. As we stated in the general response, we have made significant changes to our paper in the revision, and we hope you can take a look!

---

### Author Response · Authors · 2022-11-08
**General response to All reviewers**

Dear all reviewers,

Thank you for your time reviewing our paper! After reading the review, we realized that there exists a huge misconception of our paper. Indeed, every reviewer believed that our main contribution is using an existing algorithm to run some experiments for interpretability.

This is not our main contribution. Sorry for the misleading presentation of our original version.

Our paper focuses on a new and important dimension for interpretability, i.e., consistency. As we mentioned in Introduction, consistency is a very important requirement for many applications. Along this line, we have made the following three contributions:
1. We proved that consistency and efficiency cannot hold at the same time for interpretable algorithms.
2. We introduce a natural relaxation of efficiency called truthfulness, which is essentially the best that consistent interpretable algorithms can achieve.
3. We proved that when truthfulness is parameterized with the commonly used polynomial basis, learning truthful and consistent interpretable algorithm is equivalent to learning a Boolean function. Therefore, we propose to use classical algorithms like Harmonica and Low-degree, and observe impressive experimental results empirically.

---

We have made significant changes to our paper, and uploaded a new version with **two new theoretical results**. More specifically, our changes include:

1. **We rigorously proved a new theorem on the impossible trinity (Theorem 1, Section 3).**
2. **We rigorously proved a new lemma, stating that our Boolean functional analysis method is the unique way to generate truthful and consistent interpretations for the natural polynomial basis (Lemma 1, Section 4).**
3. We clearly stated the relationship between truthfulness and efficiency in Introduction and Section 3. Indeed, truthfulness is a natural relaxation of efficiency, especially when efficiency is not attainable for consistent interpretations.
4. We emphasized our main three contributions in our introduction.
5. We changed our abstract significantly, emphasizing our main contributions.
6. We rewrote our introduction. Now it has three different questions that we answer in our paper, corresponding to our three main contributions.
7. We add detailed discussion on why Boolean function is not a restricted function class, and was a common assumption used in many previous famous papers like LIME and SHAP.
8. We add a paragraph on previous consistent methods in Related work.
9. We moved all the existing known results to Appendix (1.5 pages!). It includes: the preliminary on Fourier analysis (Appendix A), the Harmonica algorithm and its theoretical guarantees (Appendix B), the low-degree algorithm and its theoretical guarantees (Appendix C). We are so sorry that the previous version gives the wrong impression that these results are the main contribution of our paper.
10. We moved the discussion on the existing algorithms to Appendix D, due to space limit.
11. We completely rewrite our interpretability framework in Section 3, now it should be much easier to follow, and much more rigorous. All the definitions and theorems in Section 3 are new results.
12. We added a discussion on the universal consistency at the end of Section 3, which is for the question raised by the second reviewer. We realized that without formally define consistency, it is indeed difficult to distinguish the two different notion of consistency, although the universal consistent is much more difficult to achieve.
13. We reorganized the section on learning Boolean functions, with the emphasis on our new observation of the uniqueness of our solution, as well as some discussion on why our assumptions are not strong.
14. We have added significantly more experimental results, which includes:

        a) Comparison among different algorithms for 3rd order, 2nd order and 1st order synthetic tasks.
        b) A much better visualization showing the truthful gap between our algorithm, and the others.
        c) A larger scale experiment on Imagenet, with Resnet101 and superpixels.
        d) Comparison with Harmonica and low-degree in Appendix H.
        e) Detailed experimental setting in Appendix F.

---

We also uploaded our code for the reproducibility of our experiments.

All the changes were inspired by the feedback from the reviewers, and we sincerely acknowledge your help to make our paper in a much better shape!
After so many structural changes, we do hope you could read our paper again, if you have time!

---

### Decision · Program_Chairs · 2023-01-20

**Decision:**

Reject

**Justification For Why Not Higher Score:**

Lots of issues need to be sorted out before this paper is ready for publishing.

**Justification For Why Not Lower Score:**

N/A

**Metareview: Summary, Strengths And Weaknesses:**

The paper investigates tradeoffs between consistency and simplicity. In particular, they show how the Fourier analysis of boolean functions can be used to get a sparse approximations to the target network. It was felt that the work was a bit in a restricted setting, and the authors have themselves acknowledged that the way that the paper was currently written is open to misinterpretation, in particular the focus on consistency was not apparent. Given that there has been a significant revision, it is not suitable for this paper to be accepted in this cycle.